# Product2IMG: Prompt-Free E-commerce Product Background Generation with Diffusion Model and Self-Improved LMM

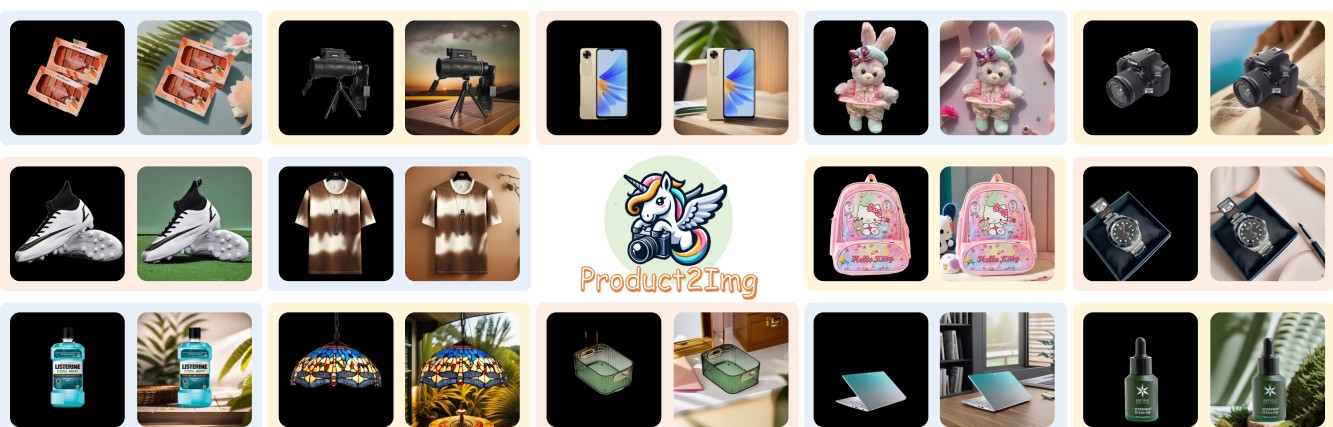

**Figure 1: Our proposed Product2Img can automatically generate corresponding backgrounds based on product images and names from various of categories, yielding high-quality results. [Best viewed in color with zoom-in].**

## ABSTRACT

In e-commerce platforms, visual content plays a pivotal role in capturing and retaining audience attention. A high-quality and aesthetically designed product background image can quickly grab consumers' attention, and increase their confidence in taking actions, such as making a purchase. Recently, diffusion models have achieved profound advancements, rendering product background generation a promising avenue for exploration. However, text-guided diffusion models require meticulously crafted prompts. The diverse range of products makes it challenging to compose prompts that result in visually appealing and semantically appropriate background scenes. Current work has made great efforts on creating prompts through expert-crafted rules or specialized fine-tuning of large language models, but it still relies on detailed human inputs and often falls short in generating desirable results by e-commerce standards.

In this paper, we propose *Product2Img*, a novel prompt-free diffusion model with automatic training data refinement strategy for product background generation. *Product2Img* employs Contrastive Background Alignment (CBA) for the text encoder to enhance the relevant background perception ability in the diffusion generation process, without the need for specific background prompts.

Meanwhile, we develop the Iterative Data Refinement with Self-improved LMM (IDR-LMM), a framework that iteratively enhances the data selection capability of LMM for diffusion model training, thereby yielding continuous performance improvements. Furthermore, we establish an E-commerce Product Background Dataset (EPBD) for the research in this paper and future work. Experimental results indicate that our approach significantly outperforms current prevalent methods in terms of automatic metrics and human evaluation, yielding improved background aesthetics and relevance.

## CCS CONCEPTS

• **Computing methodologies → Computer vision**; **Natural language processing**.

## KEYWORDS

Diffusion Model, Multimodal Learning, Background Generation

## 1 INTRODUCTION

In today's digital landscape, captivating visual content is crucial for engaging audiences, especially in e-commerce. Aesthetically pleasing, high-quality product images have the power to captivate consumers and boost confidence in their purchasing decisions. However, product images uploaded by sellers may lack professional quality, exhibiting cluttered or plain white backgrounds. This paper explores the methods to enhance product image backgrounds, aiming to elevate their contextual appeal and effectiveness.

Recently, text-to-image (T2I) diffusion models [16, 28, 31, 36] have demonstrated the ability to produce high quality visual content with textual prompts, which serve as the guidance of creativity of image generation. In order to guide the model in generating more desired content, crafting precise prompts with detailed instructions

(e.g., objectives, style, limitations) is crucial. There has been a growing community of researchers and practitioners working on developing better prompt guidelines and techniques for generation effective prompts, in terms of generation stability, reliability and consistency [4, 14, 22, 23, 40].

However, designing prompts for e-commerce product image backgrounds is still challenging. It's hard to specify appropriate background elements in a prompt, such as lighting, scenery, styling, etc. Also, creating tailored prompts for a wide array of products to perfectly match the background with the product is impractical. In response to these issues, two categories of approaches can be considered: generating prompts with a fine-tuned language model [4], and training T2I diffusion models to adhere to specific prompts [26, 45]. The first category introduces an extra language model, and such a pipeline may result in error accumulation and discontinuities in the gradient during the intermediate prompt generation process, making joint optimization challenging. This paper focuses on the second category, training end-to-end diffusion models solely using product names as conditions, without the complex prompt generation process. In specific, we propose *Product2Img*, a background generation diffusion model featuring **C**ontrastive **B**ackground **A**lignment (**CBA**). This approach employs a contrastive learning objective to align the text encoder with the latent features of backgrounds. It eliminates the need for prompt design or engineering, making it a prompt-free method.

By focusing less on designing prompts, the quality of images becomes the key factor in determining the model's outcomes. Several e-commerce product image datasets cover tasks such as image classification [11], object detection [8] or image saliency prediction [19], but they tend to offer limited product variety and overlook the aesthetic harmony between products and their backgrounds. Moreover, annotating image to meet e-commerce standards requires expertise and is labor-intensive. Recent research [1, 18, 49] highlights the effectiveness of large multimodal models (LMMs) in understanding images, suggesting a strong potential for image evaluation. However, LMMs' performance heavily depends on the specificity of their prompts, which makes it difficult for humans to include all necessary details for precise evaluations.

Inspired by previous research [25, 39] that improve LLMs through feedback and prompt refinement, we design **I**terative **D**ata **R**efinement with Self-improved **LMM** (**IDR-LMM**) framework. It employs the LMM as an effective scorer to autonomously select images that meet e-commerce standards. To the best of our knowledge, this is the first implementation of self-improvement in an LMM for continuous data refinement during model training.

To train and evaluate our method, we create **E**-commerce **P**roduct **B**ackground **D**ataset (**EPBD**), a high-quality aesthetic dataset with 25k product images with contextual background and corresponding product names. We conduct extensive experiments on EPBD, which demonstrates the effectiveness of our proposed method. We will release the dataset to facilitate further research.

Contributions of this paper are summarized as follows:

- We present *Product2Img*, a **prompt-free** method that employs Contrastive Background Alignment to generate appropriate backgrounds using only the product names, eliminating the need for complex prompting.

- We design annotation-free **IDR-LMM**, an iterative data refinement strategy, to iteratively refine training data by improving the LMM through its own feedback on diffusion model outputs. This approach effectively enhances the quality of generation results.
- We develop the high-quality E-commerce Product Background Dataset (EPBD). By releasing this dataset, we aim to encourage further research in this area.
- We achieve superior product background generation for e-commerce with an effective end-to-end approach, ensuring both aesthetic and semantic compatibility.

## 2 RELATED WORKS

### 2.1 Text-to-Image Diffusion

Text-to-Image Diffusion is a multi-modal task of generating images conditioned on texts. In the early years, popular image generation networks were mainly based on Generative Adversarial Network (GAN) [12]. Recently, diffusion models [16, 35], such as DALLE-2 [30], Imagen [33], and Stable Diffusion [31] have achieved remarkable results. Text-to-image diffusion typically involves using text encoders, such as pretrained language models like CLIP [29], to encode text inputs into latent vectors. These vectors act as conditioning signals for the diffusion model, enabling the generation of images related to the text through a process of progressively removing the noise [31].

### 2.2 Background Generation

A common operation to generate backgrounds is combining the foreground (object) from one image with another background image to create a composite image [27]. Over the past years, researchers have focused on enhancing the realism of composite images by addressing aspects such as color harmony, lighting congruence, texture matching, and geometric alignment [6, 9, 38, 42]. As diffusion models are widely applied, Paint-by-Example [43] tackles this semantic image composition problem with an image-conditioned diffusion model trained in a self-supervised manner, where the reference image is semantically transformed and harmonized before blending into another image. However, these methods all require a background image as an input to serve as a reference.

Image inpainting [2, 24, 37] emerges as a viable method for automated background generation, especially with the support of high-quality diffusion models like SD-Inpainting [31], which significantly advance image inpainting technology by fine-tuning large-scale text-to-image pre-trained model. ControlNet Inpainting [48] offers image inpainting controlled through stable diffusion with additional control signals. PowerPaint [50] achieves better text-image context alignment by learning specific task prompts, leading to cutting-edge results in context-aware image inpainting and text-guided object inpainting. However, these techniques, while impressive, do not specifically focus on background generation and require detailed prompt engineering for precise descriptions.

### 2.3 Self-Improved LLMs

"Self-improved" refers to the enhancement of LLMs using their own generated data. Research [3] has shown that large language

models (LLMs) can be improved through careful prompt engineering and techniques such as In-context Learning [10] or Chain of Thought [41]. This is because a well-crafted context can effectively elicit the desired response from LLMs. More research [7, 13, 25, 44, 46] further explores the self-improvement capability of LLMs by gathering experience from past interactions and integrating it into future queries, thereby enhancing their performance. Recent work has shown that this capability allows LLMs to exhibit significant potential in improved learning mechanisms [17] and data annotation [47]. In this paper, we extend the concept of self-improvement to LMMs, enabling them to better select training data for diffusion model.

## 3 DATASET

### 3.1 Data Collection

For the purpose of this study, we carefully curate a dataset of product images complemented by product names. To ensure image quality, 4 professional product image designers, diverse in age and gender, are hired to create high-quality images from scratch , adhering to a set of specific criteria. These criteria include the aesthetic appeal of the product image background, the relevance of the background to the product, and the avoidance of marketing text and logos within the product background.

The dataset covers a wide range of categories, ensuring its comprehensiveness. To prepare the image input for our model, we employ an in-house image cutout tool to extract the product object from each image. Then we standardize the resolution of all images to 512x512 pixels. This professionally curated collection is named E-commerce Product Background Dataset (EPBD), serving as a crucial resource for our ongoing and future research projects.

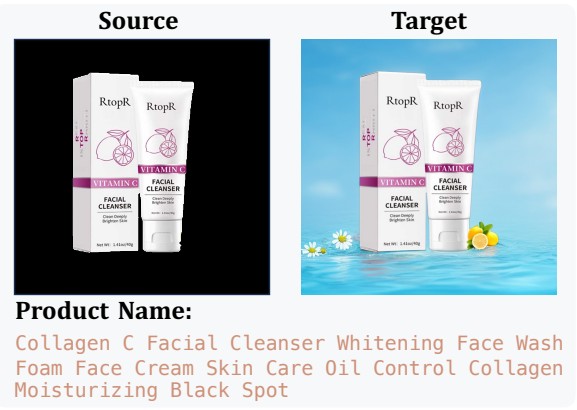

**Figure 2: Example from EPBD.**

### 3.2 Data Analysis

EPBD comprises of 25,594 samples, each including a cutout of the product, product name in English, and an attractive product image featured a contextually relevant background. For analysis and model training purposes, this dataset is stratified into specific subsets: 24,394 samples for training, 200 for validation, and 1,000 samples

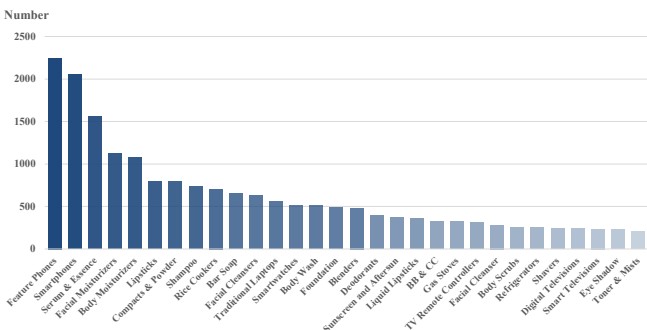

**Figure 3: Distribution of Top30 categories in EPBD.**

designated for the test set. Additionally, it contains 258 categories, the distribution of Top30 categories is graphically depicted in Figure 3. The vast range of categories underscore the complexity of training models to match an extensive array of products with varied backgrounds. Moreover, to thoroughly test the model's generalization abilities, the test set has been augmented with an extra 1,000 samples from categories beyond those in the training set, offering a more diverse and challenging evaluation scenario.

## 4 METHOD

In this section, we first introduce the task definition and preliminaries on text-to-image diffusion models. Then, we explain the motivation and design behind our proposed *Product2Img* model in detail, which is demonstrated in Figure 4 and Figure 5.

### 4.1 Task Definition

The task centers on the generation of contextually coherent and visually appealing backgrounds for product images, known as Product Background Generation. The input for this task comprises a set of product cutouts $C$ and their corresponding product names $P$. Product cutouts refer to the images of products that have been segmented from their original backgrounds, the product names serve to provide additional semantic information, aiding in the comprehension of the product types and potential usage scenarios. The goal is to create output images $I$ that seamlessly integrate these cutouts into high-quality backgrounds, tailored to the product's intended environmental setting and consumer expectations, thereby enhancing the image's suitability for e-commerce.

### 4.2 Preliminaries

Diffusion models are a class of generative models that include two processes: the diffusion process (also known as the forward process) and the denoising process (reverse process) [16]. During the forward process, noise $\epsilon \sim \mathcal{N}(\mathbf{0}, \mathbf{I})$ is added to the clean image $x_0$ as follows:

$$x_t = \sqrt{\bar{\alpha}_t} x_0 + \sqrt{1 - \bar{\alpha}_t} \epsilon, \tag{1}$$

where $x_t$ is the noisy image at time step $t$, and $\bar{\alpha}_t$ indicates the corresponding noise level.

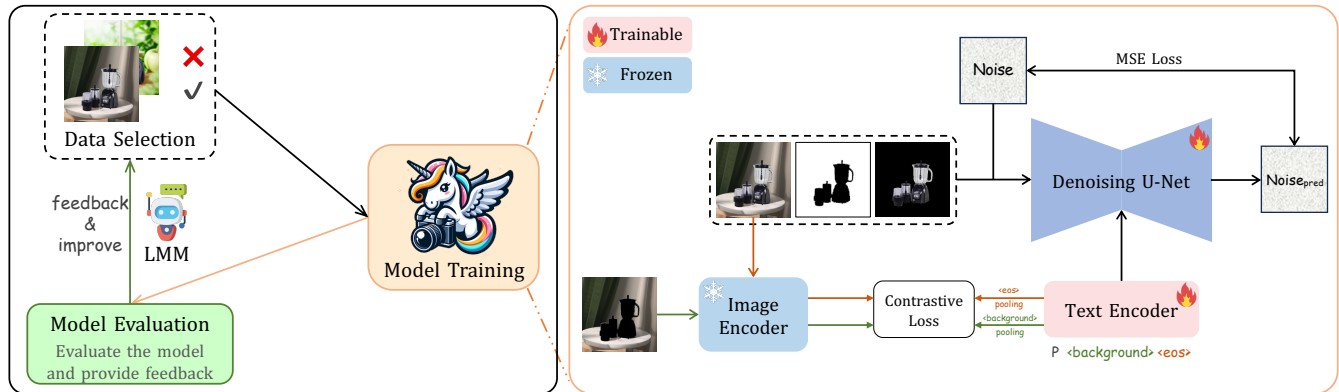

**Figure 4: Overview of the proposed *Product2Img*'s training framework. (Left) The training pipeline of the IDR-LMM. (Right) Overview of the model training for each round, $P$ refers to the product name.**

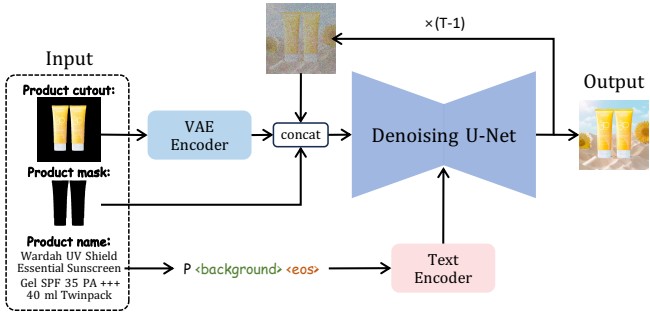

**Figure 5: The inference process of *Product2Img*.**

Correspondingly, the denoising process involves a neural network U-Net [32], parameterized by $\theta_1$, denoted as $\epsilon_{\theta_1}$, which predicts the noise to generate an image from Gaussian noise. Furthermore, diffusion models can also be conditioned on other inputs [31]. For example, text-to-image diffusion models use text features encoded by a text encoder $\Gamma_{\theta_2}$ to guide the diffusion process in the form of cross-attention, and are optimized using the following simplified objective function:

$$\mathcal{L}_{\text{simple}} = \mathbb{E}_{x_0,p,t,\epsilon} \left\| \epsilon - \epsilon_{\theta_1}(x_t, \Gamma_{\theta_2}(p), t) \right\|^2, \quad (2)$$

where $p$ represents the text conditioned on the text, i.e., the prompt.

### 4.3 Contrastive Background Alignment

CLIP [29], a multimodal model trained on the large-scale WebImageText dataset through contrastive learning, is able to understand the connections between text and images. However, the text in the dataset commonly overlook background details, making it challenging for CLIP to accurately align text with background elements. To further facilitate the comprehension of text-background matching, we devise a Contrastive Background Alignment (CBA) algorithm, which specifically learns the association between product names and the latent states of their corresponding backgrounds , thereby guiding the image diffusion model to generate appropriate background.

To achieve this, we introduce a learnable background prompt $< background >$, a special token denoted as $b$, which is initialized with the text feature vector corresponding to the term "background". It will be appended to the end of the product name $P$ to serve as the input for the CLIP text encoder. The hidden states outputted by the text encoder are then utilized as textual guidance conditions for the denoising U-Net. To fine-tune the diffusion model for background generation, we follow SD-Inpainting [31] by expanding the first convolutional layer of the denoising U-Net $\epsilon_{\theta_1}$ to include five additional channels specifically designed for the product cutout $C$ and its corresponding mask $m$. The input is composed of a noisy image latent space representation at timestep $t$, the latent space representation of the product cutout, and the mask, concatenated together and denoted as $x'_t$.

$$\mathcal{L}_{\text{inpaint}} = \mathbb{E}_{C,m,I,P,b,t,\epsilon} \left\| \epsilon - \epsilon_{\theta_1}(x'_t, \Gamma_{\theta_2}(P, b), t) \right\|^2 \quad (3)$$

To further enhance the product-background matching capability of the text encoder, we propose an auxiliary task based on contrastive learning to capture and leverage the latent correspondence between product names $P$ and product backgrounds $B$. Specifically, we treat the product name and its corresponding product background as positive samples, while pairing the product names with backgrounds of other products within the same training batch as negative samples. This approach involves minimizing the following loss function:

$$\mathcal{L}_{\text{cl}}(b, B) = -log \frac{e^{\text{sim}(\mathbf{r}_b, \mathbf{r}_B^+)/\tau}}{e^{\text{sim}(\mathbf{r}_b, \mathbf{r}_B^+)/\tau} + \sum_i e^{\text{sim}(\mathbf{r}_b, \mathbf{r}_{B_i}^-)/\tau}}, \quad (4)$$

$\mathbf{r}_b$ is the feature encoded by the text encoder and pooled through the background prompt $b$, and $\mathbf{r}_B$ denotes the image encoding features of the background image, extracted through a CLIP image encoder [29]. The term $\text{sim}(\mathbf{r}_1, \mathbf{r}_2)$ refers to the cosine similarity, which is defined as $\text{sim}(\mathbf{r}_1, \mathbf{r}_2) = \frac{\mathbf{r}_1^\top \mathbf{r}_2}{\|\mathbf{r}_1\| \cdot \|\mathbf{r}_2\|}$. Furthermore, $\tau$ is the temperature hyperparameter, which is set by default to 0.05.

Given the symmetry inherent in text-image contrastive learning, we define the symmetrized contrastive loss between the background prompt $b$ and background image $B$ as $\mathcal{L}'_{\text{cl}}(b, B) = \frac{\mathcal{L}_{\text{cl}}(b,B) + \mathcal{L}_{\text{cl}}(B,b)}{2}$.

Additionally, following CLIP [29], we also engage in contrastive learning between the product name $P$ and the image $I$ with symmetrized contrastive loss, denoted as $\mathcal{L}'_{cl}(P, I)$. Here, we use the output from the highest layer of the text encoder at the position of <eos> token as the feature of the entire sentence.

The final loss function comprises the aforementioned multiple loss components, with the specific computation process as follows:

$$\mathcal{L} = \mathcal{L}_{inpaint} + \lambda(\mathcal{L}'_{cl}(b, B) + \mathcal{L}'_{cl}(P, I)) \qquad (5)$$

where $\lambda$ represents a hyperparameter to balance the contributions of the different loss terms. To simplify the training process, we assign the same weight on contrastive loss terms. It is noteworthy that performance might see further improvement through weight adjustment.

This algorithm not only prevents catastrophic forgetting in the CLIP text encoder but also adapts the text encoder for the e-commerce domain.

## 4.4 Iterative Data Refinement with Self-improved LMM

To obtain high-quality training data that can ensure better alignment with CBA, as well as enhancing the model training, we propose an Iterative Data Refinement with Self-improved LMM (IDR-LMM). Utilizing a self-improving mechanism, the LMM can iteratively select premium training data. The whole process comprises three steps, data selection, model training, and model evaluation, as depicted in Algorithm 1.

---

**Algorithm 1** The Algorithm of IDR-LMM.

1: **Input:** The training set $\mathcal{D}^0 = \{(C_j, P_j, I_j) | j \in 1, 2, \ldots, N\}$ and the validation set $\mathcal{D}_{dev}$ of the EPBD dataset; a learnable inpainting model $\mathcal{M}_{S^0}$ and an LMM. Current iteration round $i = 1$ and maximum number of iteration rounds $\mathcal{R}$. Initial feedback $f^0$ is empty.
2: **while** The iteration $i \leq \mathcal{R}$ **do**
3:     // *Data Selection*
4:     Using feedback $f^{i-1}$ to enhance the selection of the high-quality data subset $S^i$ from $\mathcal{D}^{i-1}$ by Eq.(8)
5:     $\mathcal{D}^i \leftarrow S^i$
6:     // *Model Training*
7:     $\mathcal{M}_{S^i} \leftarrow \text{Train}(\mathcal{M}_{S^{i-1}}, S^i)$ by Eq.(5)
8:     // *Model Evaluation*
9:     $\mathbf{I}_{dev} \leftarrow \text{Infer}(\mathcal{M}_{S^i}, \mathcal{D}_{dev})$
10:     Use $\mathbf{I}_{dev}$ to obtain feedback $f^i$ to improve LMM by Eq.(7)
11:     $i \leftarrow i + 1$
12: **end while**
13: **return** $\mathcal{M}_{S^{\mathcal{R}}}$

---

4.4.1 *Data Selection.* In this approach, a dataset $\mathcal{D}$ is first defined, which consists of triplets $x = \{C, P, I\}$, representing product cutouts, product names, and product images with background. Then, a pre-trained text-to-image inpainting model $\mathcal{M} = \{\epsilon_{\theta_1}, \Gamma_{\theta_2}\}$ is fine-tuned on $\mathcal{D}$, noted as $\mathcal{M}_{\mathcal{D}}$. Our target is to select a subset $S \subset \mathcal{D}$ from $\mathcal{D}$, so that the new model $\mathcal{M}_S$ trained on $S$ can achieve improved performance compared to $\mathcal{M}_{\mathcal{D}}$.

In specific, we use the industry-leading large multimodal model GPT4-Vision [1] as an automatic scorer in our IDR-LMM method. This scorer assigns a score $R(x, p_R, c_R)$ to each sample $x \in \mathcal{D}$ using preset prompt words $p_R$ and selected samples $c_R$, with the accuracy of scoring improved by adopting techniques similar to Chain of Thought (CoT) [41] and In-context Learning (ICL) [3]. [1] Finally, samples $x_i$ whose scores are higher than a certain threshold $\gamma$ will be added to $S$, resulting in:

$$S^i \triangleq \left\{ x \in \mathcal{D}^i : R(x, p_R, c_R) \geq \gamma \right\}, \qquad (6)$$

where $i$ refers to the current iteration round, and update $\mathcal{D}^i \leftarrow S^i$.

4.4.2 *Model Training.* After selecting the subset $S$, the image inpainting model is fine-tuned. Specifically, the model is initialized with the previous iteration of the model training result $\mathcal{M}_{S^{i-1}}$ on data subset $S^{i-1}$, and a more accurate image inpainting model $\mathcal{M}_{S^i}$ is obtained by continuously minimizing the loss function 5. Particularly, $\mathcal{M}_{S^0}$ refers to the initialized model.

4.4.3 *Model Evaluation.* The models $\mathcal{M}_{S^i}$ obtained from each iteration of training are evaluated on the validation set $\mathcal{D}_{dev}$, the inference results $I_{dev}$ of the validation set are submitted to the large multimodal model for evaluation, and all evaluations are summarized to obtain feedback $f^i$.

$$f^i = \text{SUM}(F(x_{dev}, I_{dev}, p_F), p_S) \qquad (7)$$

$F(\cdot)$ represents the feedback for a single sample obtained by using prompt $p_F$ to query the GPT4-Vision, $\mathbf{x}_{dev}$ refers model inputs in the validation set, $\text{SUM}(\cdot)$ represents the LMM summarizing all feedback with prompt $p_S$.

Next, feedback $f^i$ will be incorporated into the data scoring process to help select data more suitable for training. To be specific, we incorporate feedback on the generated images from diffusion model into the LMM query, enabling LMM to be more focused on the issues in model generation. The action process of an LMM can be considered as a Partially Observable Markov Decision Process [5]. In this framework, the training and evaluation of the diffusion model, viewed as the environment, yield feedback as observations. These observed feedback, articulated in natural language, informs the LMM's subsequent actions, thereby facilitating self-improvement.

Thus the data selection equation 6 is updated to:

$$S^i \triangleq \left\{ x \in \mathcal{D}^i : R(x, p_R, c_R, f^{i-1}) \geq \gamma \right\} \qquad (8)$$

Through continuous iteration of these three steps until reaching the predetermined maximum number of iterations, this method can achieve continuous improvement in data quality and steady enhancement of model performance.

## 5 EXPERIMENT

### 5.1 Experimental Setup

5.1.1 *Implementation Details.* Our model initializes with the SD-Inpainting [31] for a robust image prior. Loss balance coefficient $\lambda$ is set to 0.01, and data selection threshold $\gamma$ is 4. Training occurs on 4 NVIDIA Tesla A100 GPUs, limited to 3 rounds of data selection, with 3000, 1000, and 1000 training steps for the first, second, and third round respectively. Batch size is 64. The learning rate for U-Net and the embedding parameters of the $< background >$ token is set to 1e-5. The parameters in the embedding layer for other

---

[1] Additional implementation details and LMM prompts ($p_R, p_F, p_S$) can be found in the appendix, located within the supplementary material.

**Table 1: Quantitative comparison of the proposed Product2Img with other methods.**

| Method | FID ↓ | PickScore ↑ | Aesthetics ↑ | CLIP-c-I ↑ | CLIP-c-B ↑ |
|---|---|---|---|---|---|
| SD-Inpainting | 5.76 | 20.09 | 4.92 | 80.15 | - |
| ControlNet-Inpainting (canny) | 8.84 | 19.55 | 4.90 | 83.56 | 54.35 |
| IP-Adapter | 8.74 | 19.72 | 4.93 | 76.79 | - |
| PowerPaint | 5.13 | 19.82 | 4.99 | 81.27 | 56.27 |
| EPBD-FT | 2.79 | 20.17 | 5.14 | 83.72 | 56.09 |
| **Product2Img** | **2.64** | **20.24** | **5.27** | **84.09** | **57.55** |

tokens in the text encoder are frozen, and the learning rate for the remaining modules in the text encoder is set to 1e-6, with both accompanied by warmup and cosine decay. For inference, a 30-step Euler Ancestral Discrete [20] sampler is used.

*5.1.2   Baselines.* To validate the superior performance of the proposed method in the task of background generation, this section compares it with the latest, competitive, and available image inpainting methods as well as an object-driven generation approach. The methods selected for comparison and their brief introductions are as follows:

- SD-Inpainting [31]: SD-Inpainting is a model that has been fine-tuned specifically for image inpainting tasks using Stable Diffusion, trained with random masks and image captions.
- ControlNet-Inpainting (canny) [48]: Controls image inpainting using Stable Diffusion with a conditional encoder that encodes masks. To prevent the model from altering product shapes while generating product images, an additional canny ControlNet is incorporated to enhance the control and improve the results.
- IP-Adapter [45]: An efficient and lightweight adapter designed to enable image prompting capabilities in pretrained text-to-image diffusion models. Unlike other baselines, this is an object-driven approach. Product images are used as image prompts and product names as text prompts to generate competitive images.
- PowerPaint [50]: A high-quality, versatile inpainting model that has shown excellent performance. It achieves the best performance in multi-purpose image inpainting by tailored fine-tuning strategies, and multi-task learning.
- EPBD-FT: EPBD-FT is a model initialized by SD-Inpainting and fine-tuned on the our EPBD dataset for product background generation.

We establish a uniform prompt template across all baselines to ensure optimal and stable performance, detailed as follows: "{Product Name}, high-quality and appropriate background".

*5.1.3   Eval Metrics.* Our objective is generating backgrounds from product cutouts and product names. The results should be realistic, aesthetically pleasing, and the background should be sufficiently relevant to the product. To evaluate from these perspectives, we employ the following metrics to assess the quality of the generated backgrounds.

- FID (Fréchet Inception Distance) [15]: FID indicates the diversity and quality of generated images and is widely used

to assess the outcomes of image generation tasks. A smaller FID indicates better image diversity and quality.
- Aesthetics [34]: Aesthetics focuses on the aesthetic quality of images and can predict human ratings of image attractiveness on a scale from 1 to 10. A high score indicates that the image's aesthetics are recognized.
- PickScore [21]: Grounded in rich user preference data, PickScore is built upon in-depth analysis and learning from a massive corpus of text-to-image samples, capturing the nuanced differences of human aesthetics. A higher PickScore indicates that the images generated based on specific prompts are more visually appealing and attractive to the audience.
- CLIP-c-I [29]: Evaluates the relevance between the generated product image $I$ and the product subject image $c$. The higher the CLIP-c-I score, the greater the fidelity of the product in the generated product image.
- CLIP-c-B [29]: Similar to CLIP-c-I, it evaluates the relevance between the product cutout $c$ and the generated background $B$. A higher CLIP score indicates a greater relevance between the product and its background in the generated image.

## 5.2   Comparison with Existing Methods

*5.2.1   Quantitative Comparison.* To accurately evaluate the effectiveness of the proposed method, we conducted a systematic performance comparison between our approach and existing techniques on the test set of EPBD. Specific experimental results are detailed in Table 1. As the table illustrates, our proposed Product2Img method demonstrates significant improvements across all evaluation metrics, which emphatically confirms the effectiveness of our approach.

It is particularly noteworthy that even the strongest baseline "EPBD-FT", which adopted a similar fine-tuning strategy on the same training set as our method, also exhibited considerable performance improvement. This further validates the quality and practical value of the EPBD. However, in direct comparisons, our method outperformed on every performance metric. Especially in visual aesthetics (Aesthetics) and CLIP-based product-background consistency (CLIP-c-B), our method achieved improvements of 0.13 and 1.46%, respectively. These figures indicate that the CBA algorithm can successfully integrate product characteristics with background features, producing background images that more closely align with the product's style. Moreover, the iterative data refinement mechanism further optimizes the model at the data quality level, resulting in generated backgrounds that are not only more aesthetically appealing but also more harmoniously consistent in terms of product-background relevance.

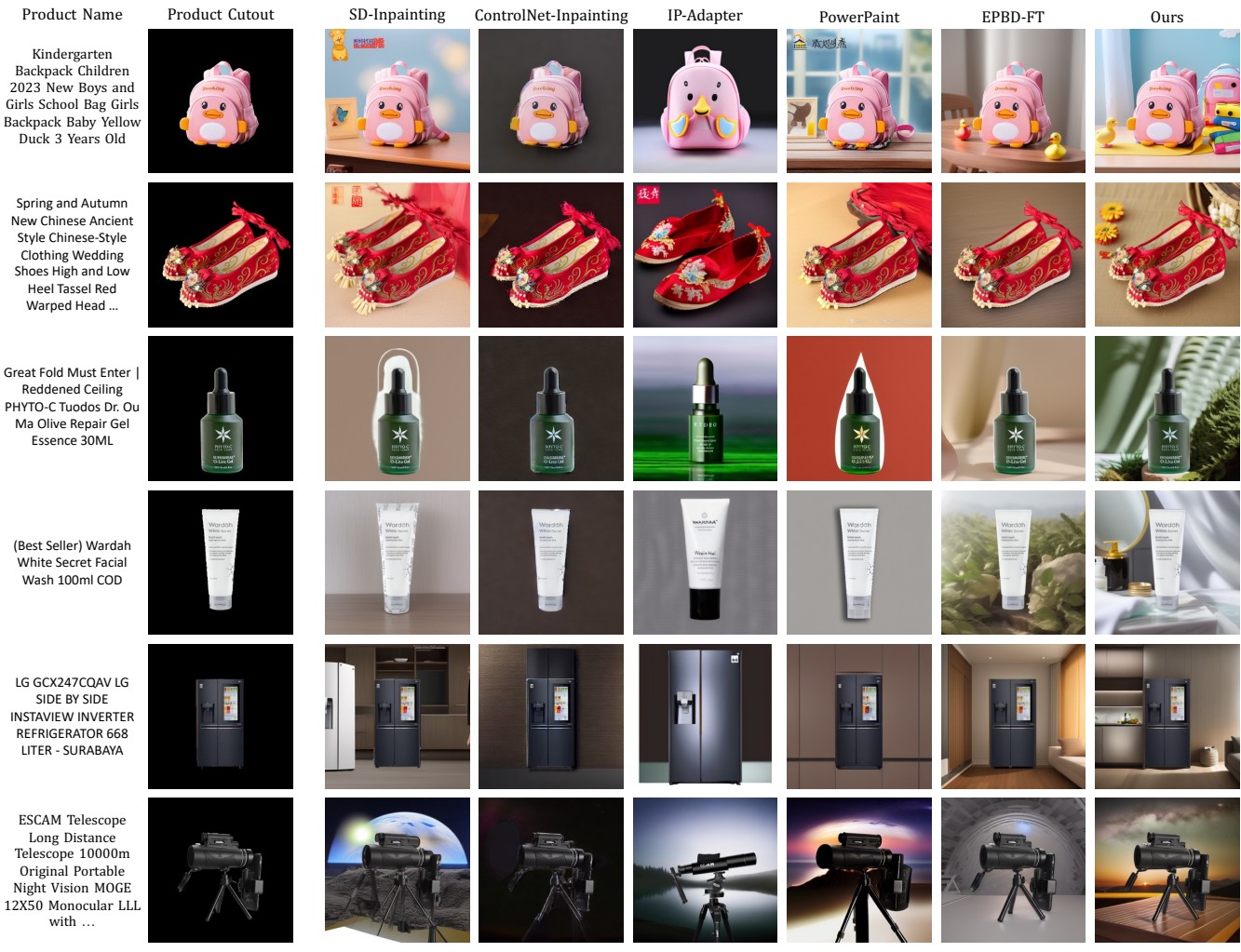

**Figure 6: Comparison of our proposed Product2Img with other methods on different kinds of products.**

*5.2.2 Qualitative Comparison.* We also choose a variety of different products for qualitative evaluation of the proposed method, covering digital products, skincare products, backpacks, apparel, etc. The qualitative comparison in Figure 6 indicates that our proposed *Product2Img* achieves state-of-the-art performance in e-commerce product background generation in terms of aesthetics, product consistency, and the match between product and background. For example, regarding the children's backpack (1st row), olive essence (3rd row), and refrigerator (5th row), our method generates corresponding backgrounds of a child-style table, a plant backdrop, and a kitchen, respectively, demonstrating a high degree of relevance and aesthetic appeal to the products. On the contrary, the EPBD-FT method tends to position these products against less relevant or solid-color backgrounds. Meanwhile, other methods also struggle with issues like subject extension, and the generation of low-quality backgrounds and watermarks.

From these cases, we also observe that our method is capable of generating a more diverse range of backgrounds which are not included in the EPBD training set, illustrating that our method can more effectively activate the background generation capabilities of the pretrained diffusion model.

*5.2.3 User Study.* To conduct a more comprehensive comparison, we further engage in user studies. Specifically, we randomly extract 200 samples from test set and invite 5 experienced professionals to select their preferred results from those generated by different methods. We randomize the presentation order of the generated results to guarantee an unbiased evaluation. The professionals are given unlimited time to make their selections based on three criteria: visual appeal, consistency between the background and the product, and overall quality.

The evaluation results are presented in Figure 7. It is evident that our method achieves the highest winning rate across all three criteria, with a significant advantage in visual appeal. In summary, this user study suggests that our method is chosen as the most satisfactory solution, demonstrating the effectiveness of our approach.

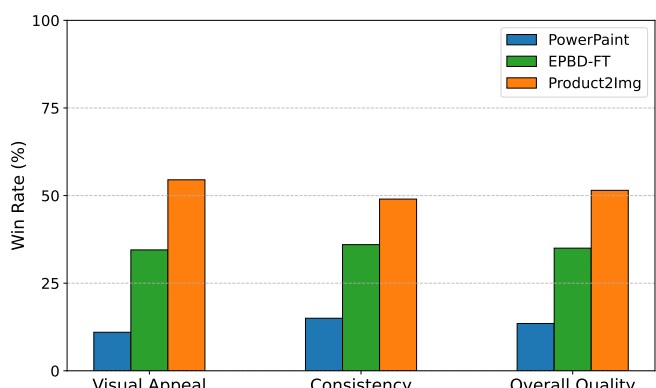

**Figure 7: User study results showing the percentage of preferred outcomes for each method based on visual appeal, background-product consistency, and overall quality.**

## 5.3 Ablation Studies and Analysis

*5.3.1 The effectiveness of CBA.* To bolster the text encoder's ability to recognize the product background details, we design CBA which integrates a contrastive learning-based auxiliary task. We conduct an ablation experiment to assess its impact by removing this task and observing performance changes. The results in Table 2 show a deterioration in performance when CBA is not applied. For example, there is a decrease from 57.30% to 56.10% in the CLIP-c-B, which underscores the effectiveness of CBA in enhancing the model's ability to generate background that are relevant to the product.

**Table 2: Ablation study For CBA (Round 1).**

|  | FID ↓ | Aesthetics ↑ | CLIP-c-B ↑ |
| --- | --- | --- | --- |
| Product2Img | **2.67** | **5.25** | **57.30** |
| w/o CBA | 2.72 | 5.24 | 56.10 |

*5.3.2 The effectiveness of IDR-LMM.* In product background generation models, data quality significantly impacts performance. We conduct experiments with IDR-LMM and thorough performance evaluations after each round. As shown in Table 3, ablation studies show that model performance steadily improves across various metrics with each round, particularly in the CLIP-c-B metric, suggesting better consistency between the generated background and the product. However, from the second to the third round, there is a slowdown in performance gain, leading us to halt at round 3. Figure 8 illustrates the improvement in training data quality across various aspects during the Iterative Data Refinement process. LMM continuously improves its data filtration capabilities by incorporating feedback obtained from evaluating images generated by the diffusion model into its queries. It is inevitable that training on progressively higher-quality data will lead to continuous improvement in model performance.

Moreover, control experiments confirm the necessity of IDR-LMM. By comparing the performance of models trained with and without IDR-LMM while maintaining consistent training steps, it

**Table 3: Experiments verifying the effectiveness of IDR-LMM on the EPBD test set, with numbers in parentheses indicating the number of preceding training steps.**

| Round | FID ↓ | Aesthetics ↑ | CLIP-c-B ↑ |
| --- | --- | --- | --- |
| Round 1 (3000) | 2.67 | 5.25 | 57.30 |
| Round 2 (4000) | **2.64** | 5.26 | 57.46 |
| Round 3 (5000) | **2.64** | **5.27** | **57.55** |
| w/o IDR-LMM (3000) | 2.68 | 5.23 | 56.28 |
| w/o IDR-LMM (4000) | 2.66 | 5.23 | 56.34 |
| w/o IDR-LMM (5000) | 2.70 | 5.23 | 56.36 |

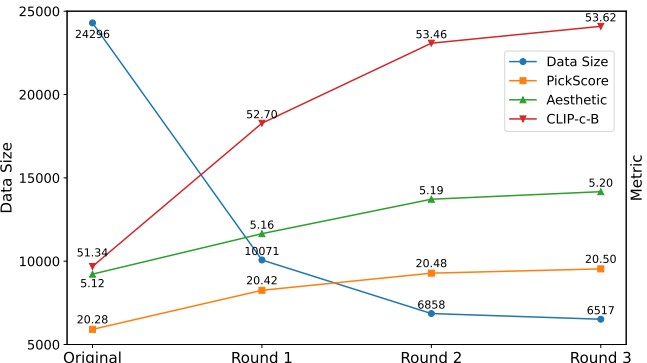

**Figure 8: The variations in data size and quality of each training round with the IDR-LMM process.**

becomes evident that data filtering not only elevates visual aesthetics but also enhances product-background matching. This indicates that the performance gains are primarily attributed to improvements in data quality, rather than mere increases in training duration.

## 6 CONCLUSION

In this paper, we propose a prompt-free *Product2Img* diffusion model for end-to-end high-quality product background generation. To accomplish this, we design a Contrastive Background Alignment (CBA) algorithm to align the hidden features between product names and their image backgrounds, thereby enhancing the relevance of the generated backgrounds and products. Additionally, we propose Iterative Data Refinement with Self-improved LMM (IDR-LMM) to gradually improve training data quality, resulting in the generation of more aesthetic and realistic backgrounds. Furthermore, we construct the E-commerce Product Background Dataset (EPBD) as the foundation for our work and future research endeavors. Comprehensive experimental evidence demonstrates that our approach substantially surpasses current prevalent techniques in generating product backgrounds with superior aesthetics and relevance. Overall, *Prodcut2Img* provides an efficient and effective approach for product backgrounds generation, fulfilling a critical demand in the e-commerce domain.

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
