# OpenReview forum: "Product2IMG: Prompt-Free E-commerce Product Background Generation with Diffusion Model and Self-Improved LMM"
_acmmm.org/ACMMM/2024/Conference — MM2024 Poster_

### Official Review · Reviewer_gmwn · 2024-05-15

**Rating:** 4
**Confidence:** 3

**Summary:**

Using generative models to design visually appealing background images for products holds immense potential for quickly capturing consumers' attention. However, the reliance on text-guided image generation poses challenges, as it necessitates careful crafting of input text, hindering background automation and adaptive generation. To address this, the paper introduces the Product2Img model, featuring an automatic training data refinement strategy tailored for product background generation sans specific background prompts. Simultaneously, the paper presents the IDR-LMM framework, aimed at iteratively enhancing the data selection ability of Large Language Models (LLMs) in training diffusion models. Additionally, the authors curate the E-commerce Product Background Dataset (EPBD) to facilitate this study and future research endeavors. Experimental results demonstrate that Product2Img surpasses current popular methods in both automatic measurement and manual evaluation. Notably, it achieves enhanced background aesthetics and relevance, showcasing its significant superiority.

**Strengths:**

1. The Product2Img model utilizes Contrastive Background Alignment to generate suitable backgrounds solely based on the product names. This innovative approach eliminates the need for textual descriptions of background design, thereby substantially reducing the overhead associated with prompt design for automated background generation.
2. The paper introduces the annotation-free IDR-LMM, an iterative data refinement strategy aimed at enhancing training data quality by iteratively improving the Large Language Model (LMM) through its feedback on diffusion model outputs. This method enables the evaluation of a high-quality subset of the dataset and facilitates continuous enhancement of model training quality. Such an approach holds significant reference value for subsequent model training endeavors.
3. The EPBD dataset introduced in this paper boasts rich classification and high image quality, representing a notable advancement in the domain of commodity background filling tasks. Its comprehensive nature and superior image quality hold significant reference value for future research endeavors in this field.

**Limitations:**

1. The IDR-LMM strategy outlined in this paper holds considerable promise for data screening and subsequent training procedures. Figure 8 visually illustrates that as training progresses, there's a noticeable reduction in the quantity of data within the subset contributing significantly to training, accompanied by a marked enhancement in the CLIP index, albeit with only marginal improvements in FID. This phenomenon prompts inquiry into whether such rapid reduction in training samples could inadvertently lead to unforeseen overfitting. Moreover, questions arise regarding the method's efficacy in adequately complementing backgrounds for Open-Domain items beyond the dataset's confines. These aspects warrant further investigation and clarification to comprehensively understand the implications of the strategy.
2. The background alignment strategy based on contrast learning, as adopted in this paper, appears to mirror the setup utilized in DreamBooth, employing special tokens to learn specific semantics, in this case, supplementary image background information.  While DreamBooth excels at learning concrete concepts, it struggles with abstract concepts such as background, a limitation that this paper attempts to mitigate to a certain extent through fine-tuning the learning constraint diffusion model.
3. Additionally, this paper introduces aesthetic indicators such as PickScore and Aesthetic.  However, the proposed method does not directly optimize image aesthetics.  As reflected in Table 2 and Table 8, there is minimal improvement in aesthetic indicators.  The perceived aesthetic superiority of Product2Img appears to stem from the high-quality EPBD dataset.  This aspect warrants further discussion and exploration.

**Suitability:**

3

---

### Official Review · Reviewer_t2Q2 · 2024-05-27

**Rating:** 5
**Confidence:** 2

**Summary:**

The paper proposes a prompt-free diffusion model with automatic training data refinement strategy for product background generation. The authors also develop an E-commerce Product Background Dataset (EPBD). Experimental results indicate that the proposed method is better than compared methods.

**Strengths:**

The paper is written well and easy to follow.
The idea of this paper is interesting, that is generating background for products.

**Limitations:**

The proposed method seems to not work for transparent products.
How to know what background is suitable for a product? From Fig.6, it's hard to know which method is better than which method.

**Suitability:**

3

---

### Official Review · Reviewer_JE5L · 2024-05-28

**Rating:** 3
**Confidence:** 3

**Summary:**

The paper presents a method to generate background images suitable for the presentation of products for e-commerce.
The idea is that often generative models are not able to capture the semantics of the background, that interplays with the semantic of the product itself.
The method doesn't require specific prompt for the background, and follows an iterative approach for training.
In addition a novel dataset is proposed.

The core elements of the proposed approach are:

-  a novel dataset (EPBD) of products/descriptions and backgrounds, curated by 4 professionals (that I expect/assume will be made available if the pair gets accepted)
- a diffusion model extended to have additional input consisting of product mask, product cut
- a training process that combines an iterative "self-improving" process (IDR-LMM) and a loss that accounts for better alignment of product image and background (CBA)

**Strengths:**

The task is interesting and quite relevant for the multimedia community.

The paper is well written, it's easy to understand the method and only minor presentation improvements may be required.

The experimental setup seems to be well designed, with ablation studies and a user study that helps to make more sense of the quantitative scores in Tab. 1. The user study shows that the images generated by the proposed approach are preferred over the competing baseline.

**Limitations:**

The experiments, as noted also by authors in Sect. 5.2, show that the dataset proposed is fundamental to obtain a good performance, since the EPBD-FT is basically a SD-Inpainting fine tuned on the dataset, with some very minor improvements when considering FD and PickScore metrics. This shows that the Contrastive Background Alignment and the iterative refinement parts of the proposed system may have a relatively reduced impact, perhaps simply increasing the size of the dataset could be enough (although the CLIP-c-I and CLIP-c-B metrics register slightly more significant improvements).
Similar tiny improvements are shown also in the ablation regarding the iterative data refinement part of the approach as shown in tab. 3, again for metrics other than CLIP-c-B, hinting that the effectiveness fo the system is more dependent on the dataset and probably on the loss that aligns CLIP with the background (CBA).

Overall the experiments show that the really important part of the system is CBA, thus reducing the novelty of the proposed system.

Improving the presentation: introduce the LMM acronym the first time it's used on page 1, not on page 2.
The terms used to describe the design of the system in Sect. 4.2 and 4.3 (e.g. \theta_1, \epsilon_theta_1, mask m, cutout C, should be used in Fig. 4 and 5, to ease parsing text and figures. Correct the typo in the name fo the system on line 925!
The bibliography should introduce some papers form the MM community. For example, in ACM MM several papers regarding generative approaches for fashion have been presented, some may be cited in this work.

**Suitability:**

3

---

### Meta-Review · Area_Chair_VL3V · 2024-06-27

**Recommendation:** Accept (Poster)
**Confidence:** 4

**Metareview:**

The reviewers agree that this is an interesting and relevant paper that is well written. The experimental setup, including ablation studies and a user study, is well designed and demonstrates the effectiveness of the proposed method. However, the reviewers mention that the novelty is rather low and the results might be heavily impacted by the dataset used for training and that the method may suffer from potential overfitting due to the rapid reduction in training samples. In this context the reviewers question the method's effectiveness for open-domain items and point out that the method does not seem to work for transparent products. However, the authors responded appropriately in their rebuttal and could clarify some of the mentioned drawbacks. In particular they show that their method does not overfit too strongly and that it is still useful for transparent product images. Considering all reviews and ratings, as well as the rebuttal of the author, I conclude that this work should be accepted.